# Swarm Control with RRT-APF Planning and FNN Task Allocation Tested on Mobile Differential Platform

**DOI:** 10.3390/s25133886

**Published:** 2025-06-22

**Authors:** Michal Lajčiak, Ján Vachálek

**Affiliations:** 1Department of Mechatronics, Secondary Technical School, 01851 Dubnica nad Vahom, Slovakia; 2Department of Applied Informatics, Automation and Mechatronics, Faculty of Mechanical Engineering, Slovak University of Technology in Bratislava, 81231 Bratislava, Slovakia

**Keywords:** swarm control, RRT APF path planning, neural network task allocation, AGV robotic agent, RMS smart manufacturing

## Abstract

This paper presents a novel method for centralized robotic swarm control that integrates path planning and task allocation subsystems. A swarm of agents is managed using various evaluation methods to assess performance. A feedforward neural network was developed to assign tasks to swarm agents in real time by predicting a suitability score. For centralized swarm planning, a hybrid algorithm combining Rapidly Exploring Random Tree (RRT) and Artificial Potential Field (APF) planners was implemented, incorporating a Multi-Agent Pathfinding (MAPF) solution to resolve simultaneous collisions at intersections. Additionally, experimental hardware using differential-drive, ArUco-tracked agents was developed to refine and demonstrate the proposed control solution. This paper specifically focuses on the swarm system design for applications in swarm reconfigurable manufacturing systems. Therefore, performance was evaluated on tasks that resemble such processes.

## 1. Introduction

A swarm of robots involves the coordination of multiple robots, or in swarm terminology, agents that can perform a collective task and solve a problem more efficiently than a single robot, defined in [1]. Real-world applications of swarm systems are rising, other than logistics where swarms are currently an industrial standard—e.g., Amazon uses more than 750,000 robots to handle logistics, as stated in [2]. Control of such robotic swarm systems mainly depends on the type of communication network utilized. Robot swarms that utilize centralized communication often rely on grid environmental models with modulations of known methods such as the Ant Colony Optimization (ACO) planning solution described in [3] or A* researched in [4]. The problem of swarms is the need for the adaptation of such planning methods for multi-agent planning without time–place collisions, known as the Multi-Agent Path Finding (MAPF) problem, researched in [5,6]. However, grid-based planning methods such as MAPF are limited in continuous and dynamic environments, where discrete grids impose unnecessary constraints on agent movement. For such free-space path planning scenarios, especially in highly reconfigurable layouts, hybrid methods like Rapidly Exploring Random Trees combined with Artificial Potential Fields (RRT-APF) offer a scalable and flexible alternative.

In contrast to the fundamental principles of Industry 4.0 described in [7], the current global focus has shifted towards the development of newer and more advanced principles, which are discussed in this article. Reconfigurable Manufacturing Systems (RMSs), originally proposed in [8], are labeled as the future of the manufacturing process. In RMSs, machines are required to change their position in real time to cover the production demand. This transportation, or reconfiguration of the entire line, can be handled by a centralized robotic swarm system as we propose and as also introduced in [9]. RMSs with swarms require positional freedom and flexibility, and they cannot be grid environments. Therefore, an effective control system for this purpose must be researched. Let us propose an example scenario for usage of an RMS with swarm—a company wants to automatize tailor-made products, but they vary in the manufacturing process due to the different varieties of the product wanted by the customers. A proper approach would be to build an RMS with swarms to handle any product variation within the factory. This type of factory would only be sustainable in the case of a high continuous demand for tailor-made products. In this research, we decided to explore the application of swarms into RMSs with the objective of developing an effective control system suitable to handle such use cases. A conceptual 3D render of a swarm applied to the reconfiguration of a manufacturing line is shown in Figure 1.

Several centralized swarm control strategies have been proposed in the literature, focusing primarily on efficient task assignment and motion coordination. Auction-based methods, such as Contract Net Protocols and First-Price Auctions, enable robots to bid for tasks based on estimated costs or utilities [10]. These strategies are often simple to implement but may result in suboptimal global allocations. More sophisticated approaches employ centralized scheduling algorithms inspired by operations research, where the allocation problem is solved as a constraint satisfaction or optimization problem [11]. Tools such as Google OR-Tools [12] provide scalable solvers for routing and scheduling problems in logistics, which can be adapted to swarm control. However, these approaches typically assume static environments or grid-based motion planning, making them less suited for dynamic or continuous-space scenarios like those found in RMSs. This research aims to overcome these limitations by combining learning-based allocation with sampling-based continuous-space planning.

## 2. Materials and Methods

### 2.1. Problem Statement

A robotic swarm is a coordinated group of *n* robots, mathematically represented as a set $S:S=\{R_{0},R_{1},\ldots ,R_{n}\}$. In this research, the robots are modeled as systems with differentially driven chassis, which results in relatively straightforward kinematics. The swarm *S* collaborates to accomplish a defined mission $M:M=\{t_{0},t_{1},\ldots ,t_{n}\}$ where each task $t_{n}$ is assigned to an individual robot $R_{n}$ or a subset $S_{n}\subset S$ consisting of fewer than *n* robots. The primary objective of deploying a robotic swarm is to accomplish the mission *M* more efficiently than a single robot operating independently. Each robot $R_{n}$ is defined by a state matrix (Equation 1) that captures its state within the environment and its structural parameters, enabling it to be treated as an object in the system.(1)$$R_{n}=\left(\begin{array}{ccccccccc} X & Y & \theta & l & w & v_{i} & \omega_{i} & E & SoC \end{array}\right)$$
where:X,Y—Cartesian coordinates of the robot,θ—Rotation around the Y-axis,l,w—Dimensions of the robot,$v_{i}$—Instantaneous velocity,$\omega_{i}$—Instantaneous angular velocity,*E*—Effector type, varying by application,SoC—State of Charge, representing the battery percentage.

The *E* parameter in (Equation 1) specifies the effector technology used by the robot, which varies based on the task requirements. For instance, $R_{0}$ may be equipped with a 3-axis robotic arm (E=1), whereas $R_{1}$ may use an electromagnetic gripper for transportation (E=2). Consequently, a task $t_{3}$ that involves machine transportation would require a robot with an electromagnetic gripper (E=2), making $R_{1}$ the suitable candidate among the available agents. Each robot $R_{n}$ is further characterized by mission-related parameters.

The swarm is considered centralized when all robots communicate with a central computer, which is responsible for generating their paths and assigning tasks. Efficient task allocation and path planning are critical for swarm performance. The system managing these functions on the central computer is referred to as the Robotic Swarm Controller, shown in Figure 2, which consists of two key subsystems: a task allocator and path planner.

The task allocation problem involves assigning a set of tasks *M* to the agents in the swarm *S* with the objective of minimizing the assignment cost. The assignment of $R_{n}$ to $t_{n}$ should represent the minimum-cost option, as opposed to assigning $R_{n+1}$.

The path planning problem can be defined as a search process for the minimum-cost path. Let $Q\subseteq R^{n}$ denote the state space of the environment, where in this research, the swarm operational environment is modeled as a Cartesian XY system without prescribed node limitations. Using such notation is needed in order to utilize swarm within modular factories with possible human presence, and also to capture to the dynamic character. Planning a path involves searching for the minimum-cost, chassis-optimized route for $R_{n}$ between the coordinates $\left\{R_{n}\left[X\right],R_{n}\left[Y\right]\right\}$ and $\left\{t_{n}\left[X\right],t_{n}\left[Y\right]\right\}$ in space *Q*, where task $t_{n}$ is assigned to robot $R_{n}$. To ensure that task $t_{n}$ is properly completed, $R_{n}$ must avoid collisions with any obstacle σ from the set of obstacles $O:O=\{\sigma_{0},\sigma_{1},\ldots ,\sigma_{n}\}$. as well as with any other robot R∈S.

An optimal path *P* is generated when it satisfies the collision-free criterion while minimizing cost, thereby increasing overall system performance. However, system effectiveness is also influenced by operational factors beyond movement, such as the effector action performed by $R_{n}$ during $t_{n}$. Both the task allocation and path planning subsystems aim to minimize cost, where the cost is defined in terms of time $\tau_{n}$ and energy $E_{n}$ consumed by $R_{n}$ to complete task $t_{n}$. The cost minimization objectives are mathematically defined in (Equation 2) and (Equation 3).(2)$$C_{t}=\min_{t}\left(\sum_{i=0}^{\left|M\right|}\tau_{i}\right)$$(3)$$C_{E}=\min_{E}\left(\sum_{i=0}^{\left|M\right|}E_{i}\right)$$

### 2.2. Performance Indicators

The performance of a robotic swarm *S* executing a mission *M* is expected to meet predefined efficiency and accuracy standards. Deviations from the expected performance, due to suboptimal path generation or task allocation, lead to a measurable decrease in the overall performance of the swarm. An ideal swarm operation would minimize such deviations, but errors are inevitable in practical scenarios. These performance losses are reflected in the increase of either the time-based cost function (Equation 2) or the energy-based cost function (Equation 3).

To quantitatively assess the performance of the swarm, appropriate Key Performance Indicators (KPIs) must be established to identify the sources of these errors. We researched our KPIs for the application of swarms specifically in modular manufacturing. Prior research [13] described most of the key metrics—scalability, robustness, fault tolerance. However, most of the KPIs are for decentralized robotic swarms. We adapted these metrics onto the centralized swarm.

#### 2.2.1. Perfomance Indicator

The swarm’s performance *P* on a mission *M* depends on key mission parameters. For instance, if the mission objective is to transport packages from place A to place B, *P* can be evaluated based on:The number of packages transported within a given time span,The amount of time required to transport a certain number of packages.

In the context of a robotic swarm factory, performance can be defined primarily by the time required to perform a reconfiguration. Since time directly correlates with production costs, it serves as the most critical factor in evaluating swarm performance within modular manufacturing systems.

#### 2.2.2. General Effectiveness Indicator

The overall effectiveness η of a swarm *S* executing a mission *M* can be expressed as the geometric mean of individual task completion rates $c_{i}$ within a unit interval:(4)$$\eta =\sqrt[{\left|M\right|}]{\prod_{i=0}^{\left|M\right|}c_{i}}$$

The task completion rate $c_{i}$ is primarily determined by the cost-time and cost-energy indicators defined in (Equation 2) and (Equation 3). The completion rate $c_{i}$ accounts for the extent to which a task is completed, such as the number of machines transported by the swarm within a mission interval compared to the expected one or any precision, e.g., placement precision parameters achieved during task execution. The geometrical mean in (Equation 4) is used to capture low completion rates and their impact on overall mission completion. The completion rate is defined as follows:(5)$$c_{i}=\frac{\max\left(1+\alpha \tau_{max}+\beta E_{max}\right)-\left(1+\alpha \tau_{i}+\beta E_{i}\right)}{\max\left(1+\alpha \tau_{max}+\beta E_{max}\right)-1}$$
where:α and β—Scaling factors that weight time and energy, respectively,$\tau_{i}$ and $E_{i}$—Time and energy consumed by robot $R_{n}$ to complete task $t_{i}$,$\tau_{max}$ and $E_{max}$—Maximum expected time and energy, obtained through testing across *n* possible path generations.

The scaling factors α and β allow for adjustment of the evaluation priority between time and energy. If minimizing time is more critical than energy consumption, then α>β, and vice versa. Equation (Equation 5) can be substituted into Equation (Equation 4), yielding a general time–energy efficiency formula for the swarm. This can be further generalized to include additional parameters $p_{l}$ that contribute to swarm effectiveness, as denoted in (Equation 6):(6)$$\eta =\sqrt[{\left|M\right|}]{\prod_{i=0}^{\left|M\right|}\frac{\max\left(1+\sum_{l=0}^{k}p_{max_{n}}\right)-\left(1+\sum_{l=0}^{k}p_{i_{n}}\right)}{\max\left(\sum_{l=0}^{k}p_{max_{n}}\right)-1}}$$
where $p_{l}$ are performance-related parameters (e.g., precision, task completion rate, or operational reliability) that contribute to the overall swarm effectiveness. This generalized form allows for evaluating swarm performance across multiple dimensions, including task accuracy, resource utilization, and operational efficiency.

Effectiveness determination can be evaluated within the time interval $\left[\tau_{0},\tau_{1}\right]$, where $\tau_{0}$ and $\tau_{1}$ represent the time constraints associated with the mission *M* or a specific task $t_{n}$. All accountable variables influencing performance are assessed within this time interval, ensuring a consistent and comprehensive evaluation of swarm efficiency. Alternatively, effectiveness can also be evaluated at an instantaneous time $\tau_{0}$ by taking the instantaneous values of the relevant parameters.

#### 2.2.3. Contribution Indicator

Imagine a scenario of swarm *S*, with, e.g., 10,000 robots working on operating an autonomous factory without breaks. The probability that $\exists R_{n}\in S;P_{R_{n}}\left(\tau_{1}\right)<P_{R_{n}}\left(\tau_{2}\right)$, where $R_{n}$ has failed before $\tau_{2}$ and is now a static obstacle at $\tau_{1}$ due to an unexpected motor failure or a collision with an obstacle or agent due to sensory error, decreases the swarm performance.

An agent $R_{n}$’s contribution to the overall η varies in τ; therefore, the failure of a certain $R_{n}$ can cause a bigger drop in one of the KPIs compared to a failure of $R_{n+1}$. Subsequently, $R_{n}$ can be replaced by a new $R_{n+2}$ from the docking station, but this process requires the transportation of $R_{n}$ to the operators to fix the issue as well as transportation of the mentioned replacement, both significantly contributing to KPI. A different case involves no agent malfunctions, where agent $R_{n}$ is assigned to $t_{n}$—contributing to overall performance—while $R_{n+1}$ is without an assigned *t* or its *t* is less valuable in *M*, e.g., a side task of collecting data to use them in analysis. In this case, *M*—the line reconfiguration—will be completed even when $R_{n+1}$ fails its task or completes it with poor KPI. This leads to a need for a weight system to analyze the contribution of any *R* to KPI. Each performance must be weighted.

#### 2.2.4. Scalability Limit Indicator

Scalability analysis examines the performance of a system as its size increases, specifically as additional units are added [14]. In the context of robotic swarms, scalability analysis evaluates the performance of a swarm *S* on a task *M* as the size of *S* increases. A swarm control system is considered scalable if its performance meets two key conditions, (Equation 7) and (Equation 8), as described in [13,15].

The performance increases or remains stable with the addition of new agents:(7)$$P_{|S|+j}\geq P_{\left|S\right|}$$The performance gain from adding new agents exceeds the average performance per agent:(8)$$P_{|S|+j}>P_{\left|S\right|}+\frac{P_{\left|S\right|}\cdot j}{\left|S\right|}$$

In centralized swarms, diminishing scalability can often be mitigated by increasing computational power and improving communication networks. Thus, scalability becomes a significant factor primarily when resources are limited. For centralized swarms, this suggests a refined definition of scalability: the performance of a robotic swarm increases only up to a certain limit, beyond which agents are no longer able to execute their paths effectively due to high robot density or high obstacle density within the operational space.

This threshold is known as the scalability limit, defined as the maximum swarm size $|S_{\max}|$ that can operate effectively within a given area. It is estimated with Formula (Equation 9):(9)$$|S_{\max}|=\frac{A\cdot (1-\rho_{o})}{a_{r}}$$
where:*A* is the total available operational area,$\rho_{o}$ is the fraction of area occupied by obstacles (obstacle area density),$a_{r}$ is the average area required per robot, including space needed for safe operation.

Note that at $|S_{d}|$ does not guarantee that the robots will be able to move safely. This condition changes dynamically, requiring an additional $a_{d}$ parameter to ensure dynamical spacing preventing deadlocks. This, added to (Equation 9), forms a new dynamic formula for maximum swarm size in given time (Equation 10):(10)$$|S_{\max}|\left(t\right)=\frac{A\cdot (1-\rho_{o})}{a_{r}+a_{d}\left(t\right)}$$

### 2.3. Task Allocation

The process of assigning a task $t_{n}$ to a robot $R_{n}$ or a group $S_{n}$ involves predicting suitability based on multiple parameters, including the state matrix, task specifications, and environmental data. Given the complexity of these non-linear relationships, a machine learning-based approach was adopted. The dynamic nature of the swarm, where |S| may vary during the execution of *M*, introduces significant challenges in network design and training. To address this, a predictive model was developed to estimate a suitability score $s_{n}$ for each robot $R_{n}$ concerning a given task $t_{n}$.

The suitability score $s_{n}$ is a normalized unit interval metric that quantifies the appropriateness of a specific robot for a particular task. The prediction process is executed for each agent, and the robot with the highest suitability score is selected for task execution. Computational efficiency is enhanced by filtering out agents that are ineligible due to insufficient battery charge, incompatible effector technology, or physical constraints that prevent successful task completion.

A Feedforward Neural Network (FNN) architecture was selected due to its demonstrated capability to model non-linear data relationships and its architectural simplicity, which facilitates efficient training and deployment. The training phase involved generating a dataset through simulation to capture the complex interactions among robot states, task requirements, and environmental variables. This simulation provided the necessary input–output pairs for training the FNN, enabling the model to generalize across diverse operational scenarios. Additionally, an alternative training approach using bootstrapped learning was explored. In this approach, a simulation is executed, and the model assigns a reward $R_{n}$ to each time step $t_{n}$. The robot follows the path, executes the task, and the effectiveness η is translated into a reward for the model. From this reward, the Mean Absolute Error (MAE) loss is computed. Therefore, using this techique, the FNN is able to generalize to heterogenous unseen scenarios as each simulation is generated. Figure 3 presents a representative plot illustrating an example of the synthetic data generation process.

The suitability score used as the output was obtained during training by evaluating the effectiveness of $R_{n}$ on task $t_{n}$. This was achieved by simulating task execution and computing the efficiency metric η using Equation (Equation 6), which is based on time τ and energy *E*. The suitability score $s_{n}$ directly reflects task execution efficiency. For example, if two robots $R_{n}$ and $R_{n+1}$ are both evaluated for a specific task $t_{n}$, and the resulting simulation yields $\eta_{n}>\eta_{n+1}$, then $R_{n}$ is deemed more suitable for executing $t_{n}$. All data was generated using a custom-built 2D continuous-space simulator developed in Python (version 3.11). The environment integrates NumPy (version 1.26) for matrix operations, Matplotlib (version 3.8) for visualization, and a basic physics engine that models robot kinematics, energy consumption, and effector–task compatibility. The simulator includes obstacle placement, collision detection, and path generation using an unoptimized RRT module. Tasks and agents were initialized with randomized positions and parameters within a bounded 1000 × 1000 unit environment under varying obstacle densities.

A Feedforward Neural Network (FNN) was developed to predict the suitability score, shown in Figure 4. The architecture consists of an input layer, a Multi Layer Perceptron (MLP) with four hidden layers, and a single output neuron activated by a sigmoid function. The network layers are defined as follows:(11)$$h^{l}=\left\{\begin{array}{ll} x & \mathrm{if}\;l=0,\\ \max(0,W^{l}h^{l-1}) & \mathrm{if}\;l=1,2,3,4,\\ \frac{1}{1+e^{-W^{5}h^{4}}} & \mathrm{if}\;l=5, \end{array}\right.$$
where in (Equation 11):l=0 (Input Layer): The input vector $x\in R^{k}$, where k=10 corresponds to the number of input features. Specifically, $x=[x_{0},x_{1},\ldots ,x_{9}]$ corresponds to the dataset variables defined in Table 1. All input features are normalized to the range [0,1] to ensure uniform scaling across the input space.l=1,2,3,4 (Hidden Layers): The architecture includes four fully connected hidden layers $h^{1},h^{2},h^{3},h^{4}$ with neuron counts K=[64,32,16,8], respectively. The ReLU activation function was selected due to its efficiency and sparse activation properties.l=5 (Output Layer): The output layer produces a single scalar value representing the suitability score for robot $R_{n}$ on task $t_{n}$. The output value is constrained to the unit interval [0,1] using a sigmoid activation function.

**Figure 4 sensors-25-03886-f004:**
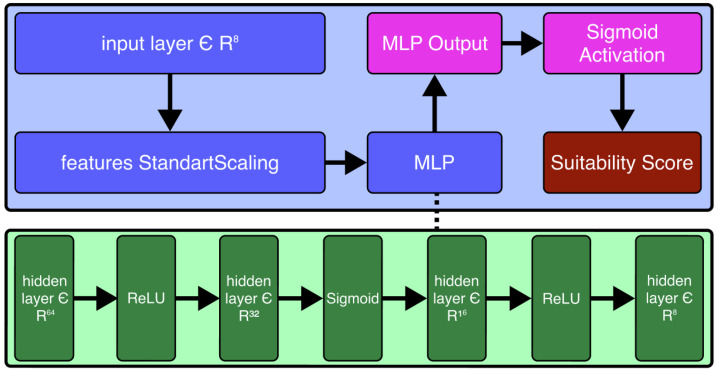
Feedforward neural network architecture diagram. The blue boxes represent preprocessing and the main multilayer perceptron (MLP) structure. Green boxes illustrate hidden layers with various activation functions. The pink box denotes the MLP output, and the red box is the final suitability score after sigmoid activation. The dashed line indicates the internal structure of the MLP.

**Table 1 sensors-25-03886-t001:** Example dataset.

Dataset Example Scenario			
Index	Data	Value	Description
0	agent-X	266.62	*X* coordinate of agent in Cartesian system within the simulation
1	agent-Y	252.20	*Y* coordinate of agent in Cartesian system within the simulation
2	agent-Effector	4	Effector technology index within the swarm equipped by agent
3	agent-Energy	7624	Energy (Joules) available in the agent’s power system (battery)
4	agent-L	100	Length of agent’s frame
5	agent-W	100	Width of agent’s frame
6	task-X	34.59	*X* coordinate of task in Cartesian system within simulation
7	task-Y	76.72	*Y* coordinate of task in Cartesian system within simulation
8	task-Effector	4	Effector technology index within the swarm required for task completion
9	distance-metric	290.91	Distance between task and agent: $\sqrt{{\left(\mathrm{agent}\text{-}X-\mathrm{task}\text{-}X\right)}^{2}+{\left(\mathrm{agent}\text{-}Y-\mathrm{task}\text{-}Y\right)}^{2}}$
10	suitability-score	0.84	Suitability score—labeled by task effectivity, calculated by Equation (Equation 6)

The ReLU activation function was chosen for the hidden layers due to its ability to mitigate the vanishing gradient problem and promote sparsity, thereby improving training efficiency.

    The network was trained using TensorFlow’s built-in Adam optimizer, which is based on a stochastic gradient descent method with adaptive moment estimation (ADAM), originally proposed in [16]. The loss function employed during training and validation was the mean absolute error (MAE). The Adam optimizer’s adaptive learning rate and moment estimation capabilities ensured fast convergence and stable learning dynamics.

### 2.4. Path Planning

Path planning algorithms are generally classified as either local or global, as described in [17]. Recent literature has explored path planning methods implemented on real-world platforms; for instance, ref. [18] proposed a cross-platform deep reinforcement learning (RL) model for autonomous local navigation, demonstrating its effectiveness across a variety of scenarios, which is highly relevant for RMS applications where flexibility and real-time adaptability are critical. Unlike grid-based methods such as ACO or A*, RL-based path planning offers robustness in continuous-space environments, which is suitable for swarm-based RMS. However, RL methods often require extensive training and may struggle with multi-agent coordination without centralized synchronization. For swarm-based operations in modular manufacturing environments, global path planning is essential. Our approach aims to develop a swarm-specific planning framework, leading to improved scalability and real-time coordination efficiency in MAPF scenarios.

#### 2.4.1. Global Path Planning

The selection of global path planning algorithms was guided by a comparative analysis of existing planning approaches, as discussed in [17,19]. In this work, multiple global path planning methods were integrated and categorized under a centralized planning framework. This is necessary because the swarm operates in a modular factory, where even local adjustments (such as collision avoidance in response to unexpected obstacles, denoted as $\sigma_{n}$) must be processed through a centralized mechanism to maintain consistency and coordination.

For path generation, we selected the Rapidly Exploring Random Tree (RRT) algorithm, originally introduced in [20]. RRT is a sampling-based method that constructs a tree of nodes in the configuration space *Q*, from which a feasible path is extracted. RRT is particularly effective for exploring non-convex and high-dimensional spaces due to its incremental nature and bias toward unexplored regions. Its capability to handle real-time scheduling requirements has been validated in [21], although the generation time is highly dependent on the complexity of *Q*, the resolution of the tree, and the swarm size |S|. However, as highlighted in [21], RRT efficiency declines significantly in environments with narrow passages or dense obstacles because a large portion of computations is consumed by collision checking.

To address RRT’s limitations in complex environments, we incorporated the Artificial Potential Field (APF) algorithm for collision avoidance and local path optimization. The APF method, first proposed in [22], defines two forces acting on the robot:$F_{rep}$—Repulsive force generated by obstacles $\sigma_{n}$ to prevent collisions,$F_{att}$—Attractive force generated by the target position $\left\{t_{n}\left[X\right],t_{n}\left[Y\right]\right\}$ to guide the robot toward the goal.

APF is well-suited for real-time dynamic adjustment because it generates smooth and continuous paths while enabling obstacle avoidance. However, it suffers from limitations such as local minima and oscillations near obstacles, which have been discussed in [23]. Despite this, APF remains valuable for improving the local efficiency of path generation, particularly when integrated with a global planning algorithm like RRT. Demonstrative plot examples of paths generated with RRT and APF are shown on Figure 5.

#### 2.4.2. Hybrid RRT-APF Approach

Hybridization of RRT and APF has been explored in recent research, where RRT generates an initial path and APF subsequently refines it for local adjustments and obstacle avoidance [24,25]. However, such hybrid approaches are often computationally expensive when scaled to multi-agent systems because they rely on post-processing of the RRT-generated path. In swarm-based settings, computational load scales with |S|, creating a bottleneck for real-time execution.

To overcome this limitation, we propose a novel inline RRT-APF hybrid approach, where APF is directly embedded into the RRT process. Specifically, for each node *r* generated by RRT, if the node collides with an obstacle $\sigma_{n}$, the APF force vector ${\vec{F}}_{APF}$ is computed and applied to relocate the node in a collision-free direction. This adjustment is formulated in (Equation 12):(12)$$\nu_{new}=\left\{\begin{array}{c} \nu_{near}\left[X\right]+\frac{step*{\vec{F}}_{APF}\left[X\right]}{∥{\vec{F}}_{APF}∥}\\ \nu_{near}\left[Y\right]+\frac{step*{\vec{F}}_{APF}\left[Y\right]}{∥{\vec{F}}_{APF}∥} \end{array}\right.$$
where:$\nu_{near}$—nearest existing node in the RRT tree,step—fixed step size for tree expansion,${\vec{F}}_{APF}$—resultant force computed from repulsive and attractive field contributions.

As RRT step size is directly related to the size of the environment as well as the obstacle density, we adopted a variable step size strategy, which is described in detail in [26]. The APF algorithm coefficients—attractive and repulsive coefficients—were tuned with an adaptive approach based on obstacles, as proposed in [27].

The inline correction using APF reduces the number of discarded nodes due to collision checking and increases the efficiency of tree expansion in complex environments. This approach ensures that RRT retains its exploratory strength while leveraging APF’s local adjustment capabilities, resulting in more direct and obstacle-aware path generation. The proposed inline hybrid approach has several advantages over conventional hybrid methods, such as the following:Reduced computational overhead—The direct correction of RRT nodes using APF reduces the need for post-processing and improves real-time performance.Improved path quality—The APF adjustment leads to smoother and more obstacle-aware paths, especially in confined environments.Reduced collision rate—Early rejection of infeasible nodes by the APF prevents excessive tree expansion in complex areas, improving the overall search efficiency.

This hybrid RRT-APF framework, described in Algorithm 1, addresses the limitations of both algorithms and enhances the swarm’s ability to execute complex pathfinding tasks in dynamic modular manufacturing settings.
**Algorithm 1** RRT-APF Path Generation  1:**for** each *i* in [0,L] **do**  2:    $\nu_{rand}=\left(x_{rand},y_{rand}\right)\leftarrow$ Randomly sampled point  3:    $\nu_{near}=\left(x_{near},y_{near}\right)\leftarrow$Nearest node in tree to $\nu_{rand}$:  4:    $\nu_{near}=\arg\min_{\nu \in T}d\left(\nu ,\nu_{rand}\right)$  5:    Generate new node in the direction of $\nu_{rand}$:  6:    Compute attractive force:  7:           $F_{att}=k_{att}\cdot \left(q_{\mathrm{goal}}-q_{\mathrm{current}}\right)$  8:    Compute repulsive force:  9:           $F_{rep}=\sum_{a=1}^{n}\frac{k_{rep}}{d_{a}^{2}}\cdot {\hat{d}}_{a}$10:    Compute total force:11:          $F_{APF}=F_{att}+F_{rep}$12:    Move $q_{\mathrm{new}}$ in the direction of $F_{total}$13:    $\nu_{new}=\nu_{near}+\frac{step*{\vec{F}}_{APF}}{∥{\vec{F}}_{APF}∥}$14:    **if** CollisionFree$(\nu_{near},\nu_{new})=\mathrm{True}$ **then**15:          Add $\nu_{new}$ to tree T16:    **else**17:          continue

Figure 6 visually demonstrates how the RRT-APF algorithm performs path planning in environments, combining the strengths of sampling-based exploration and potential field guidance. The RRT-APF path planning algorithm features multiple tunable coefficients that can significantly affect the performance of the subsystem. The variable $k_{att}$ defines the attraction towards goals, while $k_{rep}$ defines the repulsion from obstacles. When $k_{att}\gg k_{rep}$, the path may ignore obstacles, but when $k_{rep}\gg k_{att}$, the agent may get stuck, or the path will be planned inefficiently. These coefficients need to be optimized based on the environment.

#### 2.4.3. MAPF Approach

The Multi-Agent Path Finding (MAPF) system transforms path planning from a single-agent paradigm to one involving a swarm of multiple moving agents. In traditional path planning algorithms, agents compute their paths independently, without accounting for the presence of other agents. However, in the MAPF context, paths must be planned while considering potential conflicts between agents along their trajectories.

To address this, a new concept and method called the swarm clock was developed. This algorithm is named swarm clock because it tracks each agent’s position over time, enabling real-time adjustments based on interactions between agents. Swarm therefore works under a certain synchronized time—swarm clock—to prevent inconvenient planning and asynchrony. The method is described in Algorithm 2.
**Algorithm 2** MAPF Collision Prediction1:Initialize t=02:Initialize $P_{0},P_{1},\ldots ,P_{n}=\left[\right]$3:$S\leftarrow [R_{0}\leftarrow P_{0};R_{1}\leftarrow P_{1};\ldots ;R_{n}\leftarrow P_{n}]$4:**for** each $R_{n}$ in *S* **do**5:      **if** path $P_{n}$ exists **then**6:           CP←Path-Time-Intersection-Predictor(*S*)          ▹ Collision Points7:           **if** size(CP)>0 **then**8:                $P_{n}\leftarrow$Plan-Path(…, *CP*)

The Path-Time-Intersection-Predictor function takes the entire swarm state matrix *S* as input. However, in large swarm systems, the function can be optimized by considering only agents within a predefined vicinity of path $P_{n}$. The predictor calculates all possible collision points, mathematically defined in (Equation 13):(13)$$CP=P_{n}\cap \underset{i\neq n}{⋃}P_{i}$$

Here, CP denotes the set of collision points, determined by identifying intersections between path $P_{n}$ and every other path $P_{i}$, for all i≠n. These intersections are obtained by solving a system of linear Equation (Equation 14) describing the respective paths:(14)$$P_{n}\cap P_{i}\to a_{n}x+b_{n}y-c=a_{i}x+b_{i}y-c$$

Each path is represented as a linear equation. If this system yields a solution, it implies the paths intersect, signaling a potential collision. To assess whether agents are within collision range at the intersection point, a kinematic model is employed. This model defines the collision range as a circular area around the intersection with a radius equal to the agent’s physical size.

The swarm clock *t* functions as a synchronization mechanism, tracking the time elapsed since the start of the mission *M*. For instance, agent $R_{n}$ may be two-thirds through its path at time *t*, while agent $R_{n+1}$ may not have begun moving. Given the speed $v_{R_{n}}$, Cartesian position [X,Y], direction of movement θ, and trajectory $P_{n}$, the arrival time $t_{CP}$ of agent $R_{n}$ at a collision point CP can be estimated (Equation 15) (assuming the path comprises straight-line segments):(15)$$t_{CP}=t+\sum_{s=0}^{e}\frac{d_{s}}{\frac{v_{L_{i}}+v_{R_{i}}}{2}}+\frac{\Delta \theta_{i}}{\omega_{i}}$$

In this equation, *s* and *e* are the start and end indices of the path segment (from the current position to CP). The terms $v_{L}$ and $v_{R}$ denote the left and right wheel velocities of agent $R_{n}$, and the turning rate $\omega_{i}$ is given by (Equation 16):(16)$$\omega_{i}=\frac{v_{L}-v_{R}}{W}$$
where *W* is the distance between the agent’s wheels. If both $R_{n}$ and $R_{n+1}$ reach the collision point at the same time $t_{CP}$, a collision is predicted. In such cases, the algorithm regenerates the path for one or both agents to avoid the collision by introducing an imaginary obstacle at the collision point.

The swarm clock ensures synchronized movement among agents, while the collision prediction mechanism allows for dynamic, real-time path adjustments to avoid conflicts. This approach supports efficient coordination in large-scale multi-agent systems, allowing agents to operate autonomously while maintaining global harmony through continuous path updates and synchronization.

The proposed inline RRT-APF approach, combined with the swarm clock for multi-agent pathfinding, introduces several novel contributions compared to existing RRT-based path planning methods. Unlike traditional hybrid RRT-APF methods, such as those proposed by [24,25], our approach integrates APF directly into the RRT tree expansion process, enabling real-time collision avoidance during path generation rather than relying on post-processing. Ref. [24] presents APF-IRRT*, which combines an informed RRT* with APF to enhance path smoothness and convergence using a variable probability goal-bias strategy. However, their method applies APF as a post-processing step to refine RRT*-generated paths, which increases computational complexity, particularly for large swarms, due to RRT*’s iterative optimization. Similarly, ref. [25] introduce Potential Guided Directional-RRT* (PGD-RRT*), which uses potential fields to guide RRT* exploration, improving efficiency in single-agent scenarios but lacking specific mechanisms for multi-agent coordination. In contrast, our inline RRT-APF method embeds APF-based collision avoidance within the RRT tree expansion, reducing the number of discarded nodes and improving computational efficiency for both single-agent and multi-agent scenarios.

The integration of the inline RRT-APF with the swarm clock further distinguishes our approach by enabling dynamic, real-time path adjustments for multiple agents. While refs. [24,25] focus primarily on single-agent path planning or static multi-agent environments, our swarm clock synchronizes agent movements and predicts inter-agent collisions during path generation, treating collision points as imaginary obstacles within the RRT-APF framework. This ensures scalability and coordination in dynamic modular manufacturing settings, where agents must navigate complex environments with frequent interactions.

### 2.5. Experimental Hardware

The robot was created to further refine and demonstrate the researched control method. The objective was to design custom hardware without relying on existing kits, open-source projects, or commercially available robots. This approach allows the hardware to be specifically tailored to the requirements of the research, making it more adaptable to future applications in both research and educational environments. The agent was designed to meet several key objectives. First, the platform had to be low-cost, with a target production cost below EUR 60 per agent. Second, the design needed to be modular, featuring an option for swappable effectors to accommodate different experimental scenarios. Lastly, the agent was required to be max. 100×100×150 mm across dimensions for experiments to be realisable in rooms or classrooms. The design parameters were a maximum velocity of at least $0.3\;m\cdot s^{-1}$ and a numerically equal acceleration. The design had to ensure that manufacturing and assembly could be completed within an acceptable time frame, as multiple agents needed to be produced for swarm experiments, and the repairibility likely grows with assembly complexity, which is unwanted.

#### 2.5.1. Kinematic Model

The development of the platform began with the definition of its kinematic model, which directly influenced the structural design and control approach. After reviewing relevant research, it was determined that most swarm robotic platforms rely on a differential drive chassis (2WD) due to its mechanical simplicity, low cost, and compatibility with the proposed control model based on (Equation 15) and (Equation 16). Examples include HeRo 2 [28], Amir [29], and Alice [30], which successfully implemented differential drive systems for swarm applications. A differential drive system consists of two independently driven wheels, where velocity and direction are controlled by adjusting the relative speed of each wheel. Stability in such systems is provided by an additional support ball wheel (or castor) in the back of the robot [31].

The kinematic model assumes that the robot’s velocity is determined by the combined rotational speed of the two drive wheels and that friction between the wheels and the ground is negligible. Under this assumption, the velocity of the robot is computed as (Equation 17):(17)π·N·D≥v
where *D* is the wheel diameter, *v* is the desired velocity, and *N* is the wheel’s rotational speed ($s^{-1}$). Equation (Equation 17) provides a lower bound on the wheel diameter necessary to achieve the target velocity, considering the limitations of the motor’s rotational speed.

#### 2.5.2. Structural Design

The choice of motor and wheel size was guided by the kinematic model and performance requirements. Geared DC motors were selected for actuation due to their high torque-to-size ratio and cost efficiency. After market analysis, the N20 motor series from TT Motor (Shenzhen) Industrial Co., Ltd. (Shenzhen, China) was identified as the most suitable option based on cost, performance, and availability. A motor with a nominal voltage of 12 V was selected, as higher voltage increases torque output according to the DC motor torque equation. For a differential drive system, the minimal wheel diameter *D* can be computed by incorporating Newton’s Second Law into the torque equation:(18)$$M\cdot N\geq \frac{m\cdot a\cdot v}{2\cdot \pi }$$
where *M* is the motor’s torque [N·m], *m* is the robot’s mass, and *a* and *v* are the design acceleration and velocity, respectively. Equation (Equation 18) consists of motor parameters (on the left-hand side) and design parameters (on the right-hand side, which are constant); therefore, the motor parameters can be directly substituted to determine the optimal configuration.

From the available options, a 75:1 geared DC motor with a rated speed of 400 RPM and a torque of 0.157N·m satisfied this condition. The minimal viable wheel diameter was calculated as follows:(19)$$D\geq \frac{0.3\;m\cdot s^{-1}}{\pi \cdot \frac{400}{60}}=0.0143\;m=14.3\;\mathrm{mm}$$

A wheel with a diameter of 32 mm was selected based on this calculation, as it was the best market option that fit all the requirements.

Once the motor and wheels were selected, the chassis structure was designed using computer-aided design (CAD) software—Autodesk Fusion 360 version 2.0.20981. The chassis was manufactured using fused deposition modeling (FDM) 3D printing with polylactic acid (PLA) material. PLA was selected due to its favorable strength-to-weight ratio, low cost, and ease of processing, as stated in [32]. The motors and wheels were positioned on opposite sides of the chassis, with stability provided by two support ball wheels. The design aimed to minimize weight while maintaining structural integrity.

The structural design followed a layered assembly approach. The robot’s frame consisted of a floor structure where components were stacked vertically and interconnected using 3D-printed distance pillars. The modularity of the design was further enhanced by the use of a PCB-based floor structure, modeled at this stage as copper plain boards fastened together with the 3D-printed distance pillars. The magnetic cover system was added, as it ensures that the internal components are protected while allowing easy access for maintenance and modification.

The decision was made not to include the battery within the chassis itself as described Figure 7, despite the potential improvement in stability, because it would increase the height of the robot, thereby raising the center of mass and reducing maneuverability. Each motor was mounted into the main chassis body using a bracket, tightened by a nut from other the chassis side.

The top floor design, mounted atop a set of distance pillars from PCB Floor B, features a balancing weight to prevent the robot from turning over during a lift operation.

The final design met the target specifications for cost, velocity, and structural integrity. The selected motor and wheel combination provided sufficient torque and speed, while the modular chassis allowed for easy reconfiguration and component replacement. The use of 3D-printed materials reduced production costs and allowed for rapid iteration of the design, supporting the overall goal of developing a scalable, low-cost swarm robotic platform.

#### 2.5.3. Electronic Design

We then proceed to the electronics of the robot, which are divided into two printed circuit boards. PCB Floor A features all crucial components—MCU, power system etc. PCB Floor B mainly enables the modularity with interfaces for a variety of effectors.

The selected N20 motors have a nominal voltage of 12 V. The power system needed to be suited for this; therefore, an 60 g 3-series (11.1 V) 900 mAh battery was chosen. Each motor was rated for 1.6 A of stall current and needed to be controlled bidirectionally. A TB6612FNG motor driver (Toshiba Corporation, Tokyo, Japan) with in-built H-Bridge featuring two channels was utilized. Unfortunately, each channel was only rated for 1.2 A, so an overcurrent protection in the form of a PTC fuse was incorporated. The PTC switch current was 1.3 A, but the driver could handle up to 3 A momentarily, which might have even handled a stall if it occurred only for a brief moment. However, the PTC provides more reliable protection.

As depicted in Figure 8, the 12 V from the LiPo Pack are converted into 5 V and 3.3 V to supply the robot’s instruments. L7805 (STMicroelectronics, Geneva, Switzerland), outputting 5 V rated for 1.5 A, was used along with filtering eletrolyting capacitors. AMS1117 (Advanced Monolithic Systems, Santa Clara, CA, USA), outputting 3.3 V rated for 700 mA, was added to supply the systems in need of 3.3 V. The most important component selected was the Microcontroller Unit (MCU). We decided to implement a MCU and not a microchip because the application of a sole microchip would have brought a need for the installation of additional circuits, slowing down the assembly process. Since the experimental swarm is primarily designed for centralized missions, Wi-Fi was chosen as the communication tool between the central unit and the agents. We selected Wio RP2040 (Seeed Studio, Shenzhen, China) as the MCU due to its affordability and capabilities, including a dual-core 133 MHz Cortex M0+ processor (Arm Ltd., Cambridge, UK), 28 GPIO pins, an integrated ESP8285 (Espressif Systems, Shanghai, China) for Wi-Fi communication, 2 MB of Flash, and its compact surface-mount measuring 18.0 × 28.2 × 1 mm. The robot was equipped with a BGA-mounted BMM150 magnetometer (Bosch Sensortec, Reutlingen, Germany) for directional correction, a voltage sensor based on a resistor divider, and a GP2Y0A41SK0F infrared (IR) sensor (Sharp Corporation, Osaka, Japan) for collision detection, specifically measuring the distance to obstacles in the forward direction. The N20 motors in the chassis were controlled by a TB6612FNG H-bridge driver (Toshiba Corporation, Tokyo, Japan). Additionally, an additional 5V Step-Down circuit in the PCB Floor B was used to power a potential effector with a servo actuator and an A4988 stepper motor driver to control a stepper motor. A 12 V relay was also included to enable the control of additional equipment, such as an electromagnet.

The distribution of components across the two PCBs is illustrated in Figure 9b. A GPIO connector links both boards. Based on this block diagram, we designed and developed the electronic schematics and PCB layouts. Each PCB has dimensions of 90 mm × 90 mm and is equipped with a four mounting hole pattern for distance pillar stacking.

#### 2.5.4. Effector

The application of swarm into modular manufacturing introduces the need for a lift or transporation effector directly on robot. Therefore, we decided to create a one-axis electromagnetic effector.

The rotary joint drawn in Figure 10a is actuated by a MG996R servo motor (TowerPro, Shenzhen, China) with 0.94N·m, which can lift and transport a object with a mass of 500 g. The entire construction is designed to be mounted onto a magnetic cover and connected to nearby header. The design process involved a selection of materials. As mentioned, a plan was to utilize PLA for all the 3D-Printed parts. However, based on the simulation shown in Figure 10c, we see that with the usage of a PLA part, the displacement will be equal to 4.9 mm. Therefore, we decided to manufacture the effector from PETG material, which has more tensile strength than PLA, resulting in 1.1 mm displacement if loaded by 5 N of force. An electromagnet with parameters of 12 V, 1 W power, and 3 kgf was added to grip 3D-printed objects with a metal gripping zone. A rendering of the assembled effector is shown in Figure 10b.

Agents are constructed for indoor experiments; therefore, ArUco fiducial markers [33] were selected as a low-cost precise positioning system. Each agent is featured as a 50 × 50 mm marker with a 4 × 4 inner matrix. The agents operate in an close 2D space, which is bordered by 4 markers at the corners, as depicted in Figure 11b. Corner markers utilization enables the correction of perspective, which is required for proper precise detection of agents on SMART Floor with offsetted camera system.

The parameters of the assembled AGV are presented in Table 2. The final design in form of an exploded rendering is shown in Figure 12a, and the assembled system is shown in Figure 12b.

## 3. Results

To evaluate the effectiveness of the proposed model and swarm control strategy in dynamic and obstacle-rich environments, a series of simulations were conducted, aligning directly with the research objectives outlined in the previous sections.

### 3.1. Neural Network Performance

Firstly, the performance of the Feedforward Neural Network (FNN) model, trained with the Adam optimizer, was evaluated. The model utilized an 80/20 training/validation split and was assessed using Mean Absolute Error (MAE) as the loss metric. Trained over 50 epochs, the final training MAE converged to approximately 0.002, as shown in Figure 13. The validation MAE remained consistent at 0.002 (see Figure 13), indicating strong generalization and minimal overfitting. Additionally, the real-time deep-Q learning approach showed a similar convergence trend, stabilizing around a comparable MAE value.

Interpreting the suitability score as a percentage, the MAE loss translates to a deviation of only 0.2%, affirming that the FNN produces highly accurate and reliable suitability predictions. These scores directly influenced task allocation decisions, which were further validated in controller-level evaluations. To demonstrate the applicability of this method, we conducted an experiment comparing two allocation approaches: our FNN and a permutation-based Genetic Algorithm (GA).

As shown in Figure 14, the FNN consistently outperformed the GA, achieving higher average effectiveness as a maximizer. This highlights the advantage of the FNN system over a standard GA. While the GA could potentially improve with longer optimization times due to its evolutionary nature, the FNN offers faster convergence, making it more suitable for real-time applications.

### 3.2. Controller Evaluation Under Obstacle Density Variations

The swarm controller system was next evaluated through a series of simulation scenarios designed to analyze the effects of increasing obstacle density. Representative cases are depicted in Figure 15 and Figure 16. All experiments were executed locally on a MacBook Pro M3 without external computational acceleration.

Figure 17 shows plotted data from 10 scenarios for each number of obstacles. Figure 17 shows that increasing the number of obstacles leads to a proportional increase in path generation time, consistent with rising environmental density as defined by Equations (Equation 9) and (Equation 10). Simulations were conducted within a fixed 1000×1000 unit environment, using 5 robots of 20-unit size and obstacle sizes of 30 units.

At the upper bound of 50 obstacles—each occupying approximately $900\;{\mathrm{units}}^{2}$—the obstacle density $\rho_{o}$ was calculated as follows:(20)$$\rho_{o}=\frac{50\times 900}{1,000,000}=0.045\;\mathrm{or}\;4.5\%$$

This value aligns with typical machine densities found in modular manufacturing cells. Even at the density from (Equation 20), the controller maintained real-time responsiveness, with generation time peaking at just 0.3 s. Path quality remained consistent across all obstacle configurations, with generated paths averaging 12.5% deviation from the environment’s diagonal (Euclidean distance).

### 3.3. Controller Evaluation Under Robot Density Variations

Figure 18 illustrates system behavior as the number of robots increases within a 2000×2000 unit environment (total area $A=4\times 10^{6}$ units^2^) while keeping the number of obstacles fixed at 50. For each robot count, 10 scenarios were examined to capture the behavior of system. The robot footprint was estimated at roughly 600–1000 units^2^ per robot (including dynamic safety margin $a_{d}\left(t\right)$).

Using the dynamic scalability limit in Equation (Equation 10), the theoretical upper bound of the swarm size was calculated as follows:(21)$$|S_{\max}|\left(t\right)=\frac{A\cdot (1-\rho_{o})}{a_{r}+a_{d}\left(t\right)}\approx \frac{4\times 10^{6}\cdot 0.98875}{800}\approx 4940\;\mathrm{robots}$$

In practice, however, congestion effects caused by local interactions emerged at much lower densities. Although the generation time increased gradually, the controller remained stable until a practical density threshold was exceeded; therefore, the robot count follows from (Equation 21). The deviation between the planned and Euclidean paths remained consistent at approximately 12.5% across all robot quantities.

All simulations were conducted within a static environment—all obstacles were fixed in a single location throughout the simulation. Simulations in dynamic environments were not performed, but we suggest that our system would remain feasible due to the low generation times shown in Figure 17 and Figure 18. The dynamic behavior of objects would require path regeneration if such objects remain on the planned path as the agent approaches the obstacle.

### 3.4. Path Generation Failures Due to Density

As shown in Figure 19, increasing environmental density—either via more agents or obstacles—eventually led to failed path generations. These failures emerged despite the theoretical limits suggesting sufficient spatial capacity, indicating that real-time constraints such as dynamic collision risk and path contention introduced bottlenecks earlier than static models predict. This confirms the relevance of the dynamic term $a_{d}\left(t\right)$ in Equation (Equation 10), which accounts for temporary space requirements to prevent deadlocks and ensure feasible rerouting during high-density operation.

### 3.5. Analysis of KPI Effectiveness Metric

The defined effectiveness KPI in Equation (Equation 5) includes adjustable weights for each cost component. We analyzed how the performance of swarm *S* is evaluated across a set of tasks by varying α, which weights task duration, and β, which weights the energy required to complete a task. A bootstrapped-trained model was used for task allocation in this analysis, as it allows for retraining with label biases to prioritize specific cost factors.

The simulation results are shown in Figure 20. When time (α) or energy (β) is given higher relative importance, the swarm’s overall performance decreases. This occurs because the bootstrapped feedforward neural network (FNN) was not trained to generalize under asymmetric weight conditions. It was originally trained with α=β, where it performed adequately. The degradation in performance at higher α or β values suggests that the model lacks the flexibility to adapt to cost priorities it was not explicitly trained on.

### 3.6. Experimental Agent Precision Analysis

To validate the physical performance of the robotic agent, we conducted an analysis of its kinematic precision during navigation and task execution. The experiments were conducted in a controlled indoor environment using the ArUco-based positioning system described earlier. Positional accuracy was quantified by comparing the agent’s reported pose to its true location obtained via camera-based tracking.

The robot’s trajectory was planned through predefined waypoints, and deviations from the expected paths were recorded over multiple trials. Each trial consisted of the robot navigating a path and returning to its origin, repeated ten times for statistical relevance. The average deviation from the ideal path was computed using the root mean square error (RMSE) of positional data, mathematically described in (Equation 22):(22)$$\mathrm{RMSE}=\sqrt{\frac{1}{n}\sum_{i=1}^{n}{\left(x_{i}-{\hat{x}}_{i}\right)}^{2}+{\left(y_{i}-{\hat{y}}_{i}\right)}^{2}}$$
where $\left(x_{i},y_{i}\right)$ are ground truth positions and $\left({\hat{x}}_{i},{\hat{y}}_{i}\right)$ are reported positions.

The mean RMSE across all tests was found to be approximately 4.2 mm, which is within acceptable limits for swarm-scale coordination tasks and shown for a test case in Figure 21.

Overall, the agent demonstrated consistent sub-centimeter positional precision. These results affirm that the system is sufficiently precise for modular task execution in structured swarm environments, where small individual deviations are tolerable due to swarm-level redundancy and error correction.

## 4. Discussion

The primary objective of this research was to develop an effective control system for a centralized robotic swarm tailored for deployment in Reconfigurable Manufacturing Systems (RMSs). The usage of robotic swarms in RMS environments has been proposed in recent literature, such as in [9]. The proposed swarm controller consists of two subsystems: a path planner and a task allocator. Hybrid RRT-APF planning is shown to be suitable for dynamic obstacle-rich RMS environments. Embedding APF-based local correction into the node expansion of the RRT algorithm led to computationally efficient and low-noise path generation. The multi-agent pathfinding (MAPF) problem was addressed through a time-collision following method, enabling dynamic collision avoidance between agents without relying on a fully decentralized negotiation mechanism. This simplification is beneficial for RMS applications, where latency and system predictability are of high importance. For the task allocation component, the Feedforward Neural Network (FNN) trained using the Adam optimizer achieved a MAE of 0.002. This suits the allocator system for usage in swarm applications, where predicting an agent for certain task should lead to cost minimization. Simulations confirmed that the planner and allocator modules maintained real-time performance in dynamic RMS-like environments. For example, for an obstacle density of approximately 4.5%, representative of a typical modular manufacturing cell, tasks were allocated and paths generated in under 0.3 s with a swarm of size of 10 robotic agents. Scalability tests revealed the system’s ability to maintain path quality, stability, and computational efficiency under increasing numbers of robots and obstacles, up to the threshold defined in (Equation 10). The theoretical limits of robot density were rarely achieved in practical scenarios because of crowding and interference between robots. These included increased local interference between agents and a higher risk of temporary deadlocks. This case directly shows the need for adding dynamic environmental factors, such as the time-dependent spatial requirement term $a_{d}\left(t\right)$, into the maximum number of robots. The system was further validated through physical experimentation using custom-built experimental AGV robotic agents. Each AGV agent comprises an differential chassis, modular interface, wireless connectivity, and ArUco localization. Figure 22a shows photo of an AGV unit. The agents were built to be easily reprogrammable and to support future hardware integration (e.g., different sensor arrays or manipulators), making them suitable platforms for continued research and testing. The AGVs’ movement accuracy had a mean RMSE of ∼4.2 mm when tested on various tasks, e.g., a swarm applied to transportation, as shown in Figure 22b. Transitioning the experiments from the static condition-specific simulations and indoor small-scale tests to a real factory introduces new factors. Our current tests, performed in ideal setups, did not deal with real-world factory problems. For example, communication delays may lead to collisions, which would require the usage of a different positioning system, as ArUco markers are not suitable for centralized-only positioning, where communication delays cause major real trajectory offsets compared with the planned path. Electromagnetic interference from machines is common in factories. This could lead to problems with wireless signals or sensor accuracy, affecting how our robots find their way and move precisely. To fix these, we would need to utilize better communication methods that can handle errors. We also need to test our system in simulated interference.

In summary, this research presents a comprehensive methodology for centralized swarm control within RMS contexts, validated through both simulation and hardware implementation. The hybrid RRT-APF planner and neural task allocator together demonstrate that such systems can achieve real-time responsiveness, adaptability, and high precision. Future work will explore robustness under partial failure conditions, dynamic task reassignment, and integration with higher-level production planning tools.

## Figures and Tables

**Figure 1 sensors-25-03886-f001:**
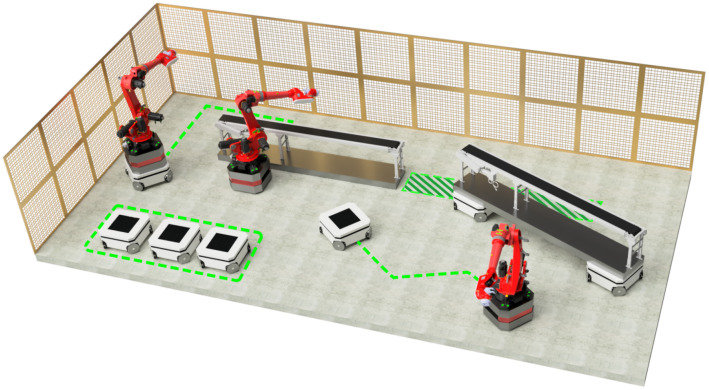
Illustrative rendering of a Reconfigurable Manufacturing System (RMS) featuring a robotic swarm application. The system includes robotic arms and mobile platforms collaborating within a flexible work environment.

**Figure 2 sensors-25-03886-f002:**
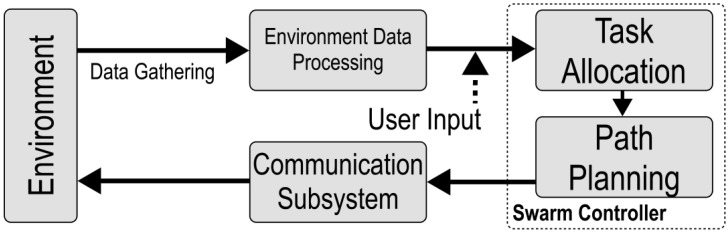
Block diagram of the swarm controller comprising task allocation and path planning subsystems.

**Figure 3 sensors-25-03886-f003:**
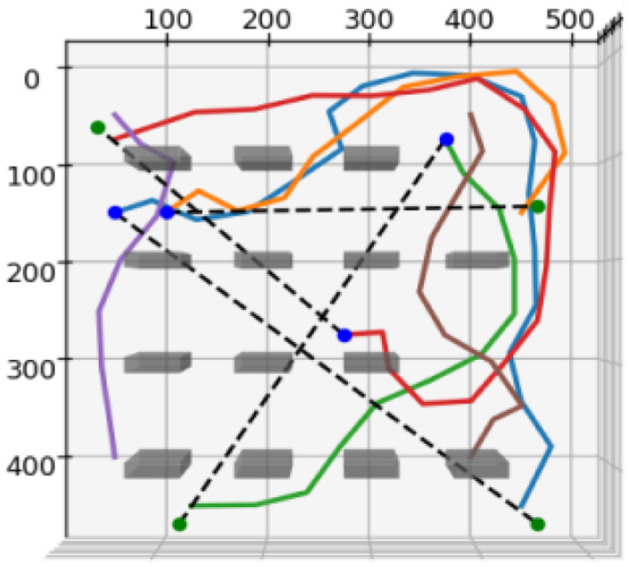
Simulation plots illustrating data collection for task and agent analysis, generated using an unoptimized RRT algorithm. (Dashed lines indicate allocation, solid lines depict paths, and colored dots represent swarm agents).

**Figure 5 sensors-25-03886-f005:**
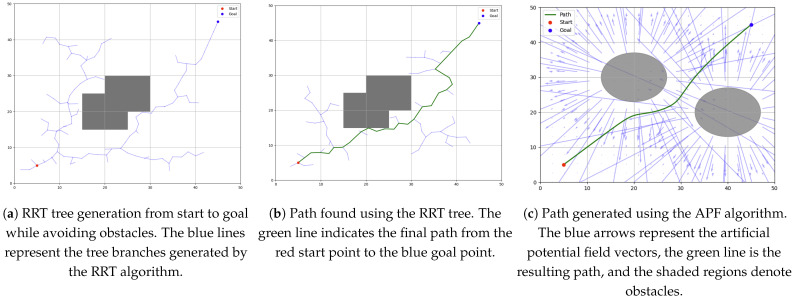
Example usage of RRT and APF algorithms for single-agent path generation. (**a**) RRT tree example. (**b**) RRT path example. (**c**) APF path planning example.

**Figure 6 sensors-25-03886-f006:**
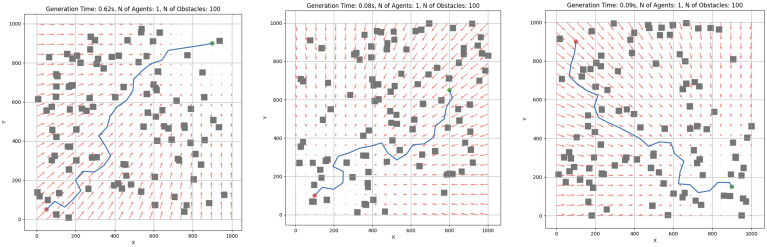
Example usage of RRT and APF algorithms for single-agent path generation. The blue line represents the path from start (red point) to goal (green point), avoiding obstacles (grey squares). Red arrows represent artificial potential field vectors (used in APF).

**Figure 7 sensors-25-03886-f007:**
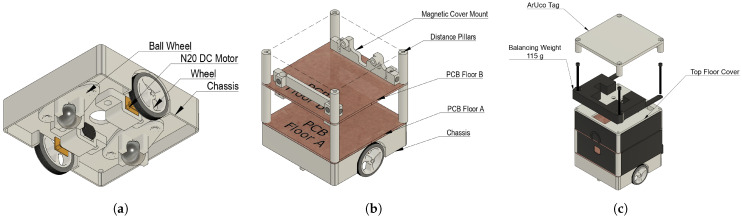
Structural design using CAD for mechanical structure of agent, manufactured by FDM 3D printing. (**a**) Chassis design. (**b**) Floor structure. (**c**) Top floor design.

**Figure 8 sensors-25-03886-f008:**
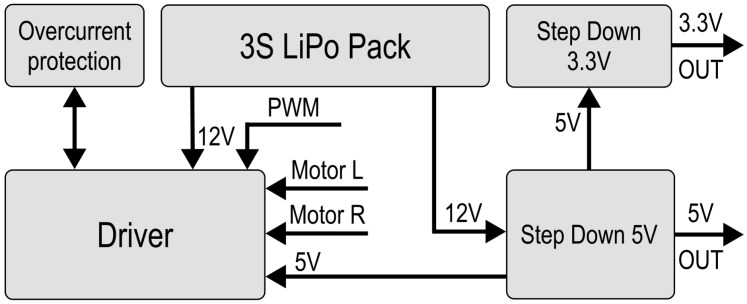
Power system block diagram.

**Figure 9 sensors-25-03886-f009:**
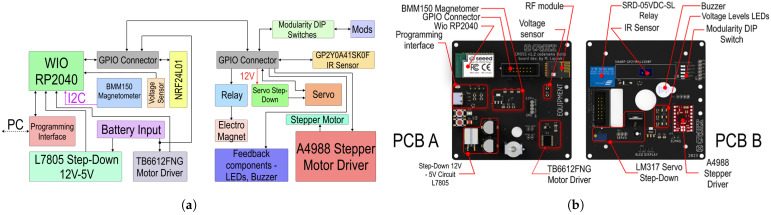
AGV robotic agent electronics. (**a**) Electronics block diagram. (**b**) PCB notation.

**Figure 10 sensors-25-03886-f010:**
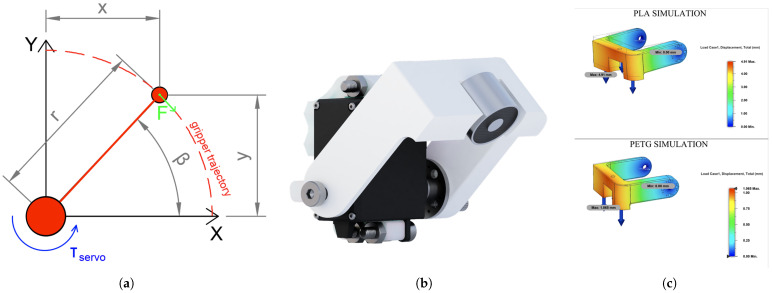
Electromagnetic servo-actuated transportation effector. (**a**) Effector diagram. (**b**) Effector rendering. (**c**) Effector’s material simulation.

**Figure 11 sensors-25-03886-f011:**
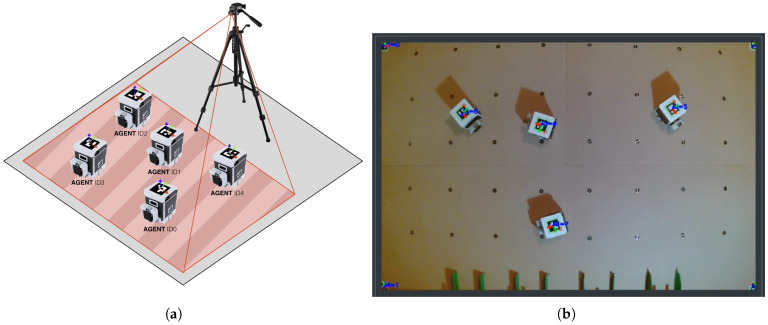
Experimental swarm localization with ArUco fiducial markers. (**a**) Detection setup. (**b**) Real tracking example of 4 agents on SMART Floor with usage of wall-mounted web camera—corners of the camera feed are incomplete as the feed is perspectevily transformed.

**Figure 12 sensors-25-03886-f012:**
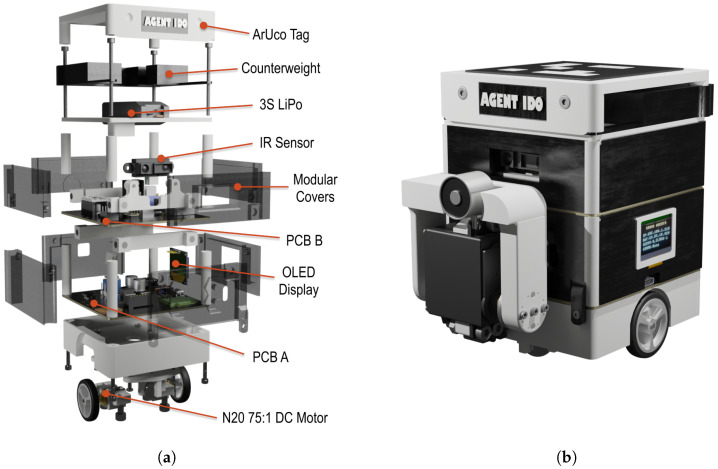
Experimental robotic agent. (**a**) Exploded agent. (**b**) Agent’s rendering.

**Figure 13 sensors-25-03886-f013:**
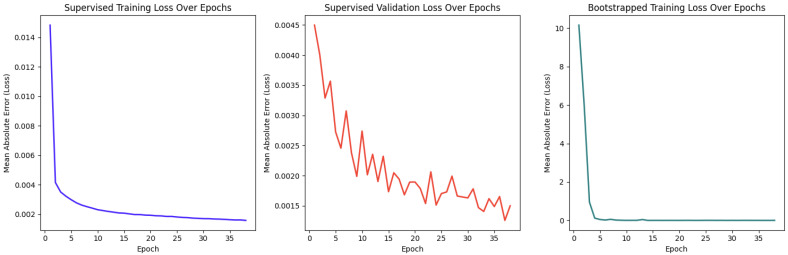
Training and validation loss over 50 epochs for the FNN model using supervised and bootstrapped learning.

**Figure 14 sensors-25-03886-f014:**
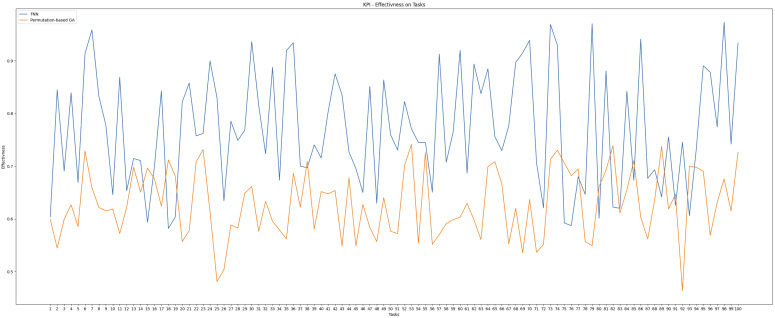
Effectiveness comparison between FNN and permutation-based GA across 100 tasks.

**Figure 15 sensors-25-03886-f015:**
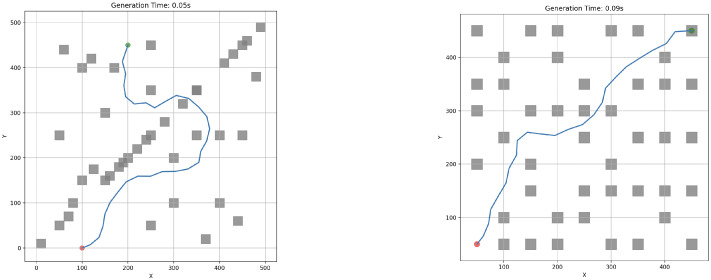
RRT-APF individual robot experimental cases. Red dot represent the starting point, where as the green dot represent the end point. Obstacles are interpreted as grey squares and RRT-APF generated path is defined by blue line.

**Figure 16 sensors-25-03886-f016:**
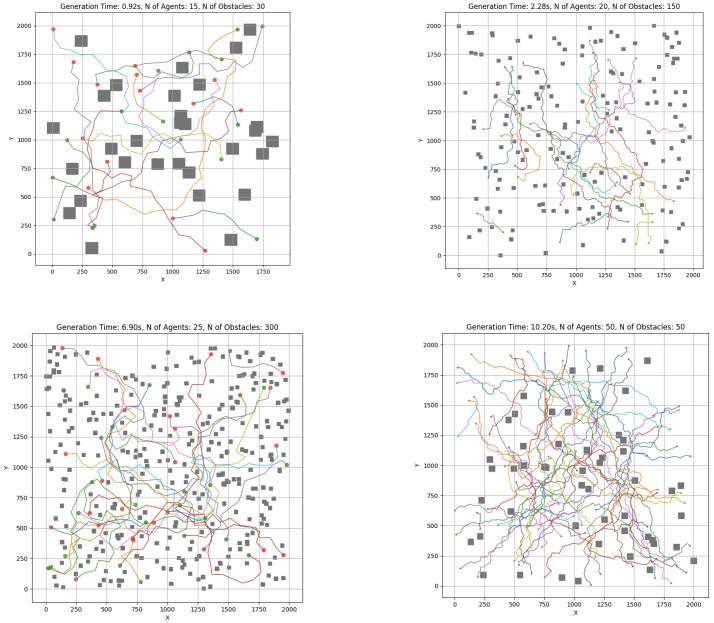
RRT-APF simulation test cases. Red dots represent indvidual starting points and green dots represent end points for each agent. Unique color of path is assigned to each agent. Grey squares represent the obstacles in each case.

**Figure 17 sensors-25-03886-f017:**
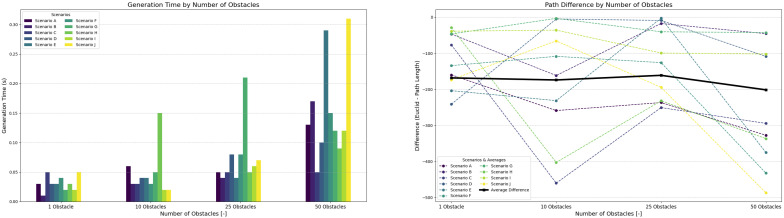
Controller performance with varying numbers of obstacles while maintaining a constant number of 10 robots per scenario.

**Figure 18 sensors-25-03886-f018:**
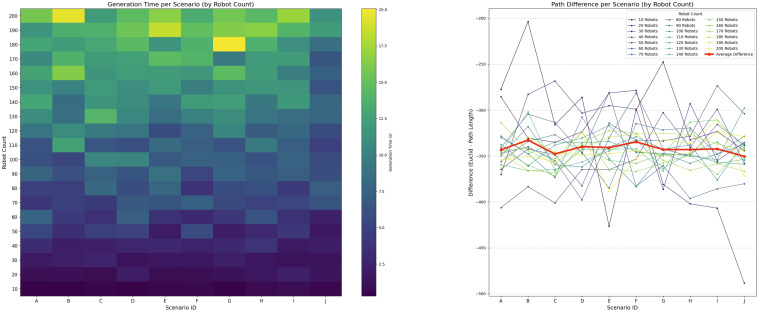
Controller performance with varying numbers of robots while maintaining a constant number of 50 obstacles per scenario.

**Figure 19 sensors-25-03886-f019:**
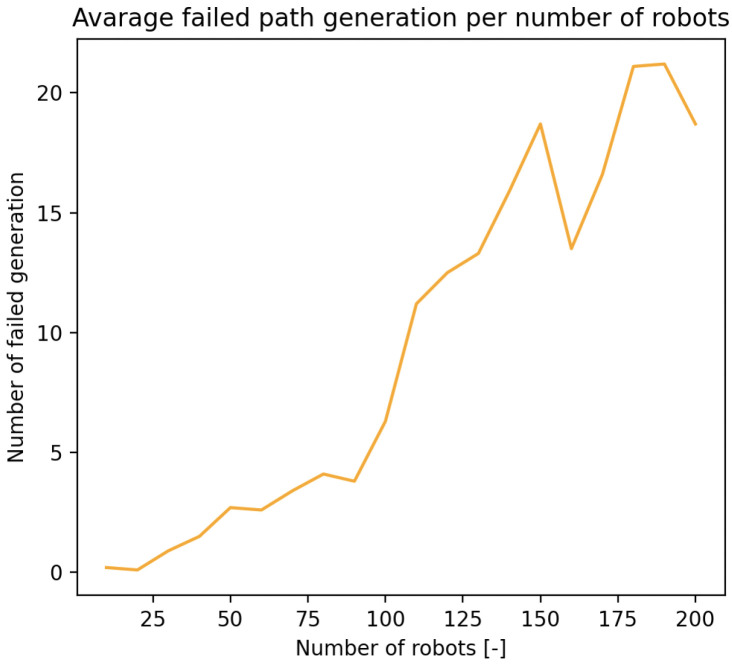
Failed path generation trend due to increasing density of the environment.

**Figure 20 sensors-25-03886-f020:**
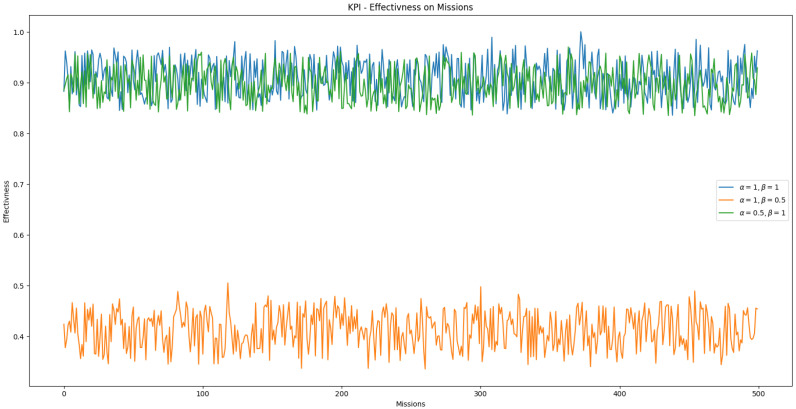
Evaluation of swarm performance on a set of missions with varying weight factors of the effectiveness KPI.

**Figure 21 sensors-25-03886-f021:**
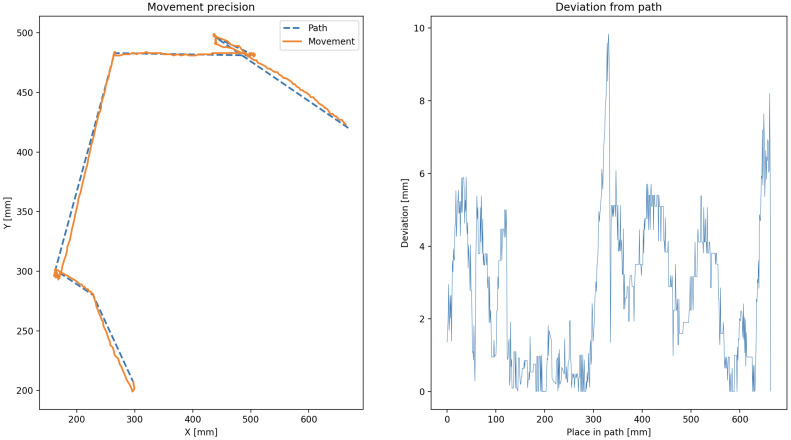
Example of planned vs. actual path with measured deviation.

**Figure 22 sensors-25-03886-f022:**
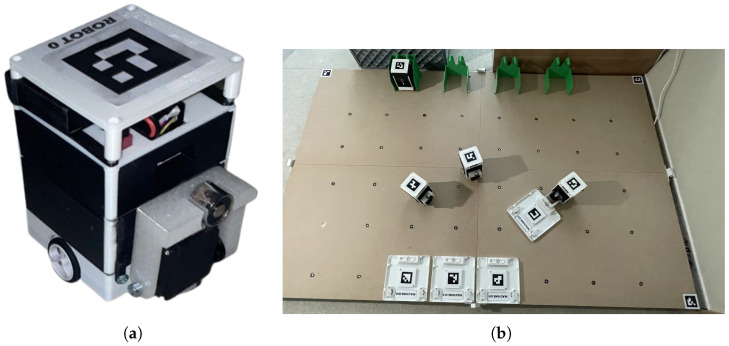
Developed AGV robotic swarm. (**a**) Photo of AGV unit. (**b**) Swarm application on SMART floor.

**Table 2 sensors-25-03886-t002:** AGV key parameters.

Experimental Robot Specifications		
Parameter	Value	Details
Dimensions	90×90×120 mm	Length × Width × Height of AGV system
Weight	450 g	Weight of the assembled robotic system with effector
Motors	DC Motor N20	Motor with 75:1 gear ratio, 400 RPM and torque of 0.157N·m
Battery	Redox LiPo	Battery with nominal 11.1 V with 3S cells and 900 mAh capacity
Battery Life	50 min	Measured value gives a time that is scalable and fully operating
Positioning System	ArUco	50 × 50 mm markers with a 4 × 4 binary matrix
Communication System	Wi-Fi	Usage of UDP to minimize time between packets
MCU	Wio RP2040	Dual-core Cortex M0+ Processor with integrated ESP8285
Max. Payload	200 g	Weight of maximum payload robot can both lift and transport

## Data Availability

The original contributions presented in this study are included in the article. Further inquiries can be directed to the corresponding authors.

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
