# Peer review of "Swarm Control with RRT-APF Planning and FNN Task Allocation Tested on Mobile Differential Platform"

_sensors, 2025, doi:10.3390/s25133886_

Round 1

Reviewer 1 Report

Comments and Suggestions for Authors

This paper proposed a swarm control approach with RRT-APF planning and FNN task allocation tested on mobile differential platform. In general, this method sounds interesting and some comments are given as follows:

  1. Although RRT-APF and FNN are combined in the text, the existing hybrid path planning methods (such as the combination of RRT* and APF) are not fully compared, and there is a lack of quantitative analysis of the unique advantages of the algorithm. Comparative experiments with classical methods need to be supplemented to highlight the innovation.
  2. The experiment was only conducted on a fixed scale (such as 10 robots and 50 obstacles) and a static environment. It did not involve dynamic obstacles or real-time task change scenarios. It is recommended to expand the dynamic environment test to verify the system robustness.
  3. The description of the robot's hardware design (such as circuit board layout and 3D printing parameters) is brief. No open-source code or detailed manufacturing process is provided, which may affect the reproduction of the experiment. Technical details and links to open-source resources need to be supplemented.
  4. The selection basis of key parameters in RRT-APF is not clear, and there is a lack of parameter sensitivity analysis. It is suggested to supplement the parameter tuning method and experimental verification.
  5. The task allocation module only relies on FNN and has not been compared with traditional methods (such as genetic algorithm). Comparative experiments need to be added to prove the advantages of FNN in terms of efficiency or accuracy.
  6. The generalization ability of FNN was evaluated only through MAE, and its performance in unseen tasks or heterogeneous robot scenarios was not discussed. Cross-scenario generalization tests and theoretical boundary analyses need to be supplemented.
  7. The experimental environment is an idealized simulation and a small-scale indoor test. Practical factors such as communication delay and electromagnetic interference in the industrial environment have not been considered. It is suggested to add a feasibility discussion on the actual factory environment.
  8. Other related path planning methods on real platforms are suggested to be discussed in the literature review part, e.g., reinforcement learning, A cross-platform deep reinforcement learning model for autonomous navigation without global information in different scenes, Control Engineering Practice 150, 105991 

Author Response

This paper proposed a swarm control approach with RRT-APF planning and FNN task allocation tested on mobile differential platform. In general, this method sounds interesting and some comments are given as follows:

Q1: Although RRT-APF and FNN are combined in the text, the existing hybrid path planning methods (such as the combination of RRT* and APF) are not fully compared, and there is a lack of quantitative analysis of the unique advantages of the algorithm. Comparative experiments with classical methods need to be supplemented to highlight the innovation.

R1: Thank you for your comment, which helped us improve the study. Based on what you have proposed, we have expanded subsection 2.4. Path Planning, specifically the MAPF Approach where we added a comparison, that clearly presents how the RRT-APF approach is unique with respect to other RRT* or APF methods. 

Q2:The experiment was only conducted on a fixed scale (such as 10 robots and 50 obstacles) and a static environment. It did not involve dynamic obstacles or real-time task change scenarios. It is recommended to expand the dynamic environment test to verify the system robustness.

R2: Thank you for your comment, due to which the study is more robust. We added an additional paragraph in the Results section in the Controller Evaluation Under Robot Density Variations subsection,  that described how our system would perform within such an environment.

Q3: The description of the robot's hardware design (such as circuit board layout and 3D printing parameters) is brief. No open-source code or detailed manufacturing process is provided, which may affect the reproduction of the experiment. Technical details and links to open-source resources need to be supplemented.

R3: Thank you for the proposal of opening the project’s details as open-source to improve the reproduction of experiments. However, currently it is not advantageous for us to do so, as we are still working on development and its application for educational purposes. To achieve this, we are communicating with partners from the commercial and industrial sphere. We think that such an approach is the best for unlocking the full potential of the hardware, and project itself. If negotiations with the patterns fail, we are planning to write another research paper, focused solely on educational Autonomous Ground Vehicles (AGVs), while releasing code and CAD as open-source. Additionally, part of this project is an award winning research presented by Michal Lajciak on world largest STEM competition ISEF 2024, where he won the 1st place award while receiving additional scholarship from foundation. Further negotiations with this foundation and other educational institutes are on the rise. However to address your proposal on point, we added a hardware summary table into section 2.5. Experimental Hardware, to help readers gain quicker and clearer perspective on hardware capabilities without a need to read the full section. Table 2. presents for instance robot’s dimensions, weight, battery, motors’ type, MCU, positioning system and key specifications.

Q4: The selection basis of key parameters in RRT-APF is not clear, and there is a lack of parameter sensitivity analysis. It is suggested to supplement the parameter tuning method and experimental verification.

R4: Your comment is well-posed and helps to improve the overall research. We used variable RRT step size described in https://doi.org/10.3390/biomimetics8040374 and variable APF coefficient tuning proposed in https://arxiv.org/abs/2504.11064 . We have added these citations in section Path Planning to add a clear perspective about the RRT-APF algorithm.

Q5: The task allocation module only relies on FNN and has not been compared with traditional methods (such as genetic algorithms). Comparative experiments need to be added to prove the advantages of FNN in terms of efficiency or accuracy.

R5: The proposal is well-posed, and we created additional simulations to further strengthen and improve the study. We created an experiment, which results are described within section 3.1. Neural Network Performance.

Q6: The generalization ability of FNN was evaluated only through MAE, and its performance in unseen tasks or heterogeneous robot scenarios was not discussed. Cross-scenario generalization tests and theoretical boundary analyses need to be supplemented.

R6: Based on your advice, we created a second training of FNN with bootstrapped learning, where FNN predicts allocation scores, allocated robots to execute task and effectiveness is propagated back as a label. Each scenario is generated and heterogeneous - each agent has different parameters. We have discussed this training method within the Results’ subsection Neural Network, where a comparative graph of loss over epoch was added..

Q7: The experimental environment is an idealized simulation and a small-scale indoor test. Practical factors such as communication delay and electromagnetic interference in the industrial environment have not been considered. It is suggested to add a feasibility discussion on the actual factory environment.

R7: Thank you for your comment. It helps in improving the overall clarity in translation of experimental setup to real manufacturing environment. Regarding this, we have added an additional section that discusses this topic in section Discussion.

Q8: Other related path planning methods on real platforms are suggested to be discussed in the literature review part, e.g., reinforcement learning, A cross-platform deep reinforcement learning model for autonomous navigation without global information in different scenes, Control Engineering Practice 150, 105991 

R8: Based on this proposal, we added a passage about deep reinforcement learning models for autonomous navigation without global information into the Material and Methods, specifically section 2.4. Path Planning, where we reviewed advantages and disadvantages of such an approach for application of MAPF into RMS. However, there is a lack of global information within the centralized system on RMS as it may struggle with multi-agent coordination. To address this, we transition to the purpose of our system - centralized, continuous environments with direct MAPF approach.

Reviewer 2 Report

Comments and Suggestions for Authors

The manuscript presents a well-executed and relevant study on centralized swarm control for reconfigurable manufacturing systems (RMS), combining an inline RRT-APF hybrid path planner and a feedforward neural network for real-time task allocation. The overall structure is solid, and the experimental platform is an outstanding contribution. However, several key areas require revision or clarification to strengthen the manuscript before it can be recommended for publication.

- The introduction sets the industrial context effectively but lacks a critical review of related centralized swarm control strategies (e.g., centralized auction-based allocation, scheduling methods, OR-tools).

- MAPF is well-cited, but the transition from traditional grid-based planning to free-space RRT-APF hybridization should be better justified.

- Please clarify the novelty with respect to previous works combining RRT and APF in multi-agent settings.

- The KPI-based effectiveness metrics are well-motivated but lack experimental sensitivity analysis. What happens if α ≫ β or vice versa?

- In task allocation, the neural network architecture is reasonable, but:

    -- Dataset generation is insufficiently described. What simulation tool/environment was used?

    -- The labeling mechanism using η based on simulations needs empirical justification.

    -- Was overfitting assessed with k-fold validation or only with one split?

- I want to personally comment on the impressive effort on building custom AGVs. However, this section would benefit from a summary table with key specs (e.g., battery, motor torque, max payload).

Author Response

The manuscript presents a well-executed and relevant study on centralized swarm control for reconfigurable manufacturing systems (RMS), combining an inline RRT-APF hybrid path planner and a feedforward neural network for real-time task allocation. The overall structure is solid, and the experimental platform is an outstanding contribution. However, several key areas require revision or clarification to strengthen the manuscript before it can be recommended for publication.

Q1: The introduction sets the industrial context effectively but lacks a critical review of related centralized swarm control strategies (e.g., centralized auction-based allocation, scheduling methods, OR-tools).

R1: Thank you for your comment, which contributes to clarify the introduction and thereby the coherence of the paper. Based on it, we have added an additional paragraph in the section Introduction discussing the importance of methods you propose and how they are used within centralized swarm allocation. We discussed the usage of Google OR-Tools for allocation subsystems along with Contract Net and First-Price Auctions. The paragraph presents, why aren’t these algorithms applicable onto our approach on centralized swarm applied to RMS.

Q2: MAPF is well-cited, but the transition from traditional grid-based planning to free-space RRT-APF hybridization should be better justified.

R2: Based on your proposal, we have added an additional paragraph into the Introduction section that justifies the transition from grid-based planning to free space planning.

Q3: Please clarify the novelty with respect to previous works combining RRT and APF in multi-agent settings.

R3: Thank you for your comment, which helped us improve the study. Based on what you have proposed, we have expanded subsection 2.4. Path Planning, specifically the MAPF Approach where we added a comparison, that clearly presents how the RRT-APF approach is unique with respect to other RRT* or APF methods. 

Q4: The KPI-based effectiveness metrics are well-motivated but lack experimental sensitivity analysis. What happens if α ≫ β or vice versa?

R4: We have created additional simulation to answer your question, which is well stated and improves the quality of our research. For this we added subsection Analysis of KPI Effectiveness Metric in section Results, where by may simulation clarify the effectiveness KPI of bootstrapped-trained FNN with tuning the KPI’s weights.

Q5: In task allocation, the neural network architecture is reasonable, but: Dataset generation is insufficiently described. What simulation tool/environment was used?

R5: Thank you for your comment. We insufficiently described the data generation process. Due to it we have added a paragraph in section Task Allocation which shows this process - clarifies the technology and describes the process.

Q6: The labeling mechanism using η based on simulations needs empirical justification.

R6: The proposal is well situated and has led to additional retraining with a different approach, which led to the additions. We incorporated bootstrapped learning based on the effectiveness, with which the robot completes tasks that the network assigns him to. This led to faster convergence to global minimum with MAE as loss, while similar loss values were achieved as within the supervised training.This new approach is described within the Results’ subsection Neural Network, where a comparative graph of loss over epoch was added.

Q7: Was overfitting assessed with k-fold validation or only with one split?

R7: The question was responded within the Results’ subsection Neural Network Performance, as we have used an one split of 80/20 (training/validation). Usage of k-fold validation split was not required, as the model showed feasible performance within the one-split training approach as we haven’t runned into overfitting issues.

Q8: I want to personally comment on the impressive effort on building custom AGVs. However, this section would benefit from a summary table with key specs (e.g., battery, motor torque, max payload).

R8: Thank you for your advice. We have added Table 2​​ AGV Key Parameters which describes key parameters and thereby helps to gain a clearer perspective about the hardware without a need to read the full section in detail, as it highlights only the key factors.

Round 2

Reviewer 1 Report

Comments and Suggestions for Authors

The authors have addressed all my comments.

Reviewer 2 Report

Comments and Suggestions for Authors

This version of the manuscript is significantly more complete and ready for publication.